# Symmetric Single Index Learning

**Aaron Zweig**
Courant Institute of Mathematical Sciences
New York University
New York, NY 10012, USA
`az831@nyu.edu`

**Joan Bruna**
Courant Institute of Mathematical Sciences
Center for Data Science
New York University
New York, NY 10012, USA
`bruna@cims.nyu.edu`

## Abstract

Few neural architectures lend themselves to provable learning with gradient based methods. One popular model is the single-index model, in which labels are produced by composing an unknown linear projection with a possibly unknown scalar link function. Learning this model with SGD is relatively well-understood, whereby the so-called information exponent of the link function governs a polynomial sample complexity rate. However, extending this analysis to deeper or more complicated architectures remains challenging.

In this work, we consider single index learning in the setting of symmetric neural networks. Under analytic assumptions on the activation and maximum degree assumptions on the link function, we prove that gradient flow recovers the hidden planted direction, represented as a finitely supported vector in the feature space of power sum polynomials. We characterize a notion of information exponent adapted to our setting that controls the efficiency of learning.

## 1 Introduction

Quantifying the advantage of neural networks over simpler learning systems remains a primary question in deep learning theory. Specifically, understanding their ability to discover relevant low-dimensional features out of high-dimensional inputs is a particularly important topic of study. One facet of the challenge is explicitly characterizing the evolution of neural network weights through gradient-based methods, owing to the nonconvexity of the optimization landscape.

The single index setting, long studied in economics and biostatistics (Radchenko, 2015) offers the simplest setting where non-linear feature learning can be characterized explicitly. In this setting, functions of the form $x \mapsto f(\langle x, \theta^* \rangle)$ where $\theta^* \in \mathcal{S}_{d-1}$ represents a hidden direction in high-dimensional space, and $f$ a certain non-linear link function, are learned via a student with an identical architecture $x \mapsto f(\langle x, \theta \rangle)$, under certain data distribution assumptions, such as Gaussian data. Gradient flow and gradient descent (Yehudai & Ohad, 2020; Arous et al., 2021; Dudeja & Hsu, 2018) in this setting can be analyzed by reducing the high-dimensional dynamics of $\theta$ to dimension-free dynamics of appropriate *summary statistics*, given in this case by the scalar correlation $\langle \theta, \theta^* \rangle$.

The efficiency of gradient methods in this setting, measured either in continuous time or independent samples, is controlled by two main properties. First, the correlation initialization, which typically scales as $\frac{1}{\sqrt{d}}$ for standard assumptions. Second, the information exponent $s_f$ of $f$ (Arous et al., 2021; Dudeja & Hsu, 2018; Bietti et al., 2022; Damian et al., 2023; 2022; Abbe et al., 2023), which measures the number of effective vanishing moments of the link function — leading to a sample complexity of the form $O(d^{s-1})$ for generic values of $s$.

While this basic setup has been extended along certain directions, e.g. relaxing the structure on the input data distribution Yehudai & Ohad (2020); Bruna et al. (2023), considering the multi-index counterpart Damian et al. (2022); Abbe et al. (2022; 2023); Arnaboldi et al. (2023), or learning the link function with semi-parametric methods Bietti et al. (2022); Mahankali et al. (2023), they are all fundamentally associated with fully-connected shallow neural networks. Such architecture, for all its rich mathematical structure, also comes with important shortcomings. In particular, it is unable to account for predefined symmetries in the target function that the learner wishes to exploit. This requires specialized neural architectures enforcing particular invariances, setting up novel technical challenges to carry out the program outlined above.

In this work, we consider arguably the easiest form of symmetry, given by permutation invariance. The primary architecture for this invariance is DeepSets (Zaheer et al., 2017), which is necessarily three layers by definition and

therefore not a simple extension of the two layer setting. In order to quantify the notion of 'symmetric' feature learning in this setting, we introduce a symmetric single index target, and analyze the ability of gradient descent over a DeepSets architecture to recover it. Under appropriate assumptions on the model, initialization and data distribution, we combine the previous analyses with tools from symmetric polynomial theory to characterize the dynamics of this learning problem. Our primary theorem is a proof of efficient learning under gradient flow, with explicit polynomial convergence rates controlled by an analogue of information exponent adapted to the symmetric setting. Combined with other contemporary works, this result solidifies the remarkable ability of gradient descent to perform feature learning under a variety of high-dimensional learning problems.

## 2 SETUP

### 2.1 NOTATION

For $z \in \mathbb{C}$, we will use $\overline{z}$ to denote the complex conjugate, with the notation $z^*$ always being reserved to denote a special value of $z$ rather than an operation. For complex matrices $A$ we will use $A^\dagger$ to denote the conjugate transpose. The standard inner product on $\mathbb{C}^N$ is written as $\langle \cdot, \cdot \rangle$, whereas inner products on $L^2(\gamma)$ spaces for some probability measure $\gamma$ will be written as $\langle \cdot, \cdot \rangle_\gamma$. Furthermore, for $h$ a vector and $p(x)$ a vector-valued function, we will use $\langle h, p \rangle_\gamma$ as shorthand for the notation $\langle h, p(\cdot) \rangle_\gamma$.

### 2.2 REGRESSION SETTING AND TEACHER FUNCTION

We consider a typical regression setting, where given samples $(x, y) \in \mathcal{X} \times \mathbb{C}$ with $y = F(x)$, we seek to learn a function $F_w$ with parameter $w \in \mathbb{C}^M$ by minimizing some expected loss $E_{x \sim \nu}[L(F(x), F_w(x))]$. Note that we consider complex-valued inputs and parameters because they greatly simplify the symmetric setting (see Proposition 2.3), hence we will also assume $\mathcal{X} \subseteq \mathbb{C}^N$. Both $F$ and $F_w$ will be permutation invariant functions, meaning that $F(x_{\pi(1)}, \ldots x_{\pi(N)}) = F(x_1, \ldots, x_N)$ for any permutation $\pi : \{1, N\} \to \{1, N\}$.

Typically the single index setting assumes that the trained architecture will exactly match the true architecture (e.g. as in Arous et al. (2021)), but below we will see why it's necessary to consider separate architectures. For that reason, we'll consider separately defining the teacher $F$ and the student $F_w$.

The first ingredient are the power sum polynomials:

**Definition 2.1.** *For $k \in \mathbb{N}$ and $x \in \mathbb{C}^N$, the* normalized powersum polynomial *is defined as*

$$p_k(x) = \frac{1}{\sqrt{k}} \sum_{n=1}^N x_n^k .$$

Let $p(x) = [p_1(x), p_2(x), \ldots]$ be an infinite dimensional vector of powersums, and consider a fixed vector $h^* \in \mathbb{C}^\infty$ of unit norm. Then our teacher function $F$ will be of the form

$$F : \mathcal{X} \to \mathbb{C} \tag{1}$$
$$x \mapsto F(x) := f(\langle h^*, p(x) \rangle) \tag{2}$$

for some scalar link function $f : \mathbb{C} \to \mathbb{C}$. $F$ may thus be understood as a single-index function in the feature space of powersum polynomials.

### 2.3 DEEPSETS STUDENT FUNCTION

Let us remind the typical structure of a DeepSets network (Zaheer et al., 2017), where for some maps $\Phi : \mathcal{X} \to \mathbb{C}^M$ and $\rho : \mathbb{C}^M \to \mathbb{C}$, the standard DeepSets architecture is of the form:

$$x \mapsto \rho(\Phi_1(x), \ldots, \Phi_M(x)) . \tag{3}$$

The essential restriction is that $\Phi$ is a permutation invariant mapping, typically of the form $\Phi_m(x) = \sum_{n=1}^N \phi_m(x_n)$ for some map $\phi_m : \mathbb{C} \to \mathbb{C}$. In order to parameterize our student network as a DeepSets model, we will make the simplest possible choices, while preserving its non-linear essence. To define our student network, we consider the symmetric embedding $\Phi$ as a one-layer neural network with no bias terms:

$$\Phi_m(x) = \sum_{n=1}^N \sigma(a_m x_n) , \tag{4}$$

for i.i.d. complex weights sampled uniformly from the complex circle $a_m \sim S^1$ and some activation $\sigma : \mathbb{C} \to \mathbb{C}$. And given some link function $g : \mathbb{C} \to \mathbb{C}$, we'll consider the mapping $\rho$ as:

$$\rho_w(\cdot) = g(\langle w, \cdot \rangle) , \tag{5}$$

where $w \in \mathbb{C}^M$ are our trainable weights. Putting all together, our student network thus becomes

$$\begin{aligned} F_w &: \mathcal{X} \to \mathbb{C} \\ x &\mapsto F_w(x) := g(\langle w, \Phi(x) \rangle) . \end{aligned} \tag{6}$$

Explicitly, this corresponds to a DeepSets network with only one trainable vector $w$, while the first layer weights $\{a_m\}_{m=1}^M$ and the third layer weights that parameterize the function $g$ are frozen.

The first fact we need is that, through simple algebra, the student may be rewritten in the form of a single-index model.

**Proposition 2.2.** *There is matrix $A \in \mathbb{C}^{\infty \times M}$ depending only on the activation $\sigma$ and the frozen weights $\{a_m\}_{m=1}^M$ such that*

$$g(\langle w, \Phi(x) \rangle) = g(\langle Aw, p(x) \rangle) . \tag{7}$$

## 2.4 Hermite-like Identity

In the vanilla single index setting, the key to giving an explicit expression for the expected loss (for Gaussian inputs) is a well-known identity of Hermite polynomials (O'Donnell, 2021; Jacobsen, 1996). If $h_k$ denotes the Hermite polynomial of degree $k$, this identity takes the form

$$\langle h_k(\langle \cdot, u \rangle), h_l(\langle \cdot, v \rangle) \rangle_{\gamma_n} = \delta_{kl} k! \langle u, v \rangle^k , \tag{8}$$

where $u, v \in \mathbb{R}^n$ and $\gamma_n$ is the standard Gaussian distribution on $n$ dimensions.

In our setting, as it turns out, one can establish an analogous identity, by considering a different input probability measure, and a bound on the degree of the link function. We will choose our input domain $\mathcal{X} = (S^1)^N$, and the input distribution we will consider is the set of eigenvalues of a Haar-distributed unitary matrix in dimension $N$ (Diaconis & Shahshahani, 1994), or equivalently the squared Vandermonde density over $N$ copies of the complex unit circle (Macdonald, 1998). We'll interchangeably use the notation $\mathbb{E}_{x \sim V}[f(x)\overline{g(x)}] = \langle f, g \rangle_V$.

**Proposition 2.3.** *Consider $h, \tilde{h} \in \mathbb{C}^\infty$ with bounded $L_2$ norm. For exponents $k, l$ with $k \leq \sqrt{N}$, if $h$ is only supported on the first $\sqrt{N}$ elements, then:*

$$\langle \langle h, p \rangle^k, \langle \tilde{h}, p \rangle^l \rangle_V = \delta_{kl} k! \langle h, \tilde{h} \rangle^k . \tag{9}$$

The crucial feature of this identity is that the assumptions on support and bounded degree only apply to $\langle h, p \rangle^k$, with no restrictions on the other term. In our learning problem, we can use this property to make these assumptions on the teacher function, while requiring no bounds on the terms of the student DeepSets architecture.

In order to take advantage of the assumptions on the support of $h$ and the degree in the above proposition, we need to make the following assumptions on our teacher link function $f$ and our true direction $h^*$:

**Assumption 2.4.** *The link function $f$ is analytic and only supported on the first $\sqrt{N}$ degree monomials, i.e.*

$$f(z) = \sum_{j=1}^{\sqrt{N}} \frac{\alpha_j}{\sqrt{j!}} z^j \tag{10}$$

*Furthermore, the vector $h^*$ is only supported on the first $\sqrt{N}$ elements.*

Although this assumption is required to apply the orthogonality property for our loss function in the following sections, we note that in principle, including exponentially small terms of higher degree in $f$ or higher index in $h^*$ should have negligible effect. Moreover, one should interpret this assumption as silently disappearing in the high-dimensional regime $N \to \infty$. For simplicity, we keep this assumption to make cleaner calculations and leave the issue of these small perturbations to future work.

## 2.5 INFORMATION EXPONENT

Because Proposition 2.3 takes inner products of monomials, it alludes to a very simple characterization of information exponent. Namely:

**Definition 2.5.** *Consider an analytic function $f : \mathbb{C} \to \mathbb{C}$ that can be written in the form*

$$f(z) = \sum_{j=0}^{\infty} \frac{\alpha_j}{\sqrt{j!}} z^j \tag{11}$$

*Then the* information exponent *is defined as* $s = \inf\{j \geq 1 : \alpha_j \neq 0\}$.

Similar to the Gaussian case (Arous et al., 2021; Bietti et al., 2022), the information exponent $s$ will control the efficiency of learning. Assuming $|\alpha_s|$ is some non-negligible constant, the value of $s$ will be far more important in governing the convergence rate.

## 2.6 CHOOSING A LEARNABLE LOSS

There are two subtleties to choosing an appropriate loss function. Namely, the necessity of a correlational loss (with regularization), and the necessity of choosing the student and teacher link functions to be distinct.

At first glance, it is tempting to simply define a loss of the form

$$\tilde{L}(w) = \mathbb{E}_{x \sim V} |F(x) - F_w(x)|^2 = \mathbb{E}_{x \sim V} \left[ |f(\langle h^*, p(x) \rangle) - f(\langle Aw, p(x) \rangle)|^2 \right] . \tag{12}$$

However, the Deepsets student model is not degree limited, that is the support of $Aw$ is not restricted to the first $\sqrt{N}$ terms of the powersum expansion. In other words, expanding this loss will require calculating the term $\|f(\langle Aw, p \rangle)\|_V^2$, which will contain high degree terms that cannot be controlled with Proposition 2.3. One could avoid this issue by choosing the activation such that $Aw$ only contains low-index terms, but we want to consider larger classes of activations and enforce fewer restrictions.

One can instead consider a correlational loss. In this case, in order to make the objective have a bounded global minimum, it's necessary to either regularize $w$, or project at every step of SGD, which is the strategy taken in Damian et al. (2023). In our setting, this projection would correspond to projecting $w$ to the ellipsoid surface $\|Aw\| = 1$. This projection would require solving an optimization problem at every timestep (Pope, 2008). To avoid this impracticality, we instead consider regularization.

Then with complete knowledge of the link function $f$, specifically its monomial coefficients, we can now define the correlational loss

$$\hat{L}(w) = \mathbb{E}_{x \sim V} \left[ -\operatorname{Re}\left\{ f(\langle h^*, p(x) \rangle \overline{f(\langle Aw, p(x) \rangle)}) \right\} \right] + \sum_{i=j}^{\sqrt{N}} \frac{|\alpha_j|^2}{2} \|Aw\|^{2j} . \tag{13}$$

This loss enjoys benign optimization properties, as shown by the following proposition:

**Proposition 2.6.** *If there exist coprimes $k, l$ with $\alpha_k, \alpha_l \neq 0$, and $h^*$ is in the range of $A$, then $\hat{L}$ exclusively has global minima at all $w$ such that $Aw = h^*$.*

However, unlike the real case, complex weights causes issues for learning this objective. Namely, this objective can be written as a non-convex polynomial in $\cos\theta$ where $\theta$ is the angle of $\langle Aw, h^* \rangle$ in polar coordinates.

Therefore, we consider a different choice of student link function that will enable a simpler analysis of the dynamics. For the choice of $g(z) = \frac{\alpha_s}{|\alpha_s|\sqrt{s!}} z^s$, we instead consider the loss:

$$L(w) = \mathbb{E}_{x \sim V} \left[ -\operatorname{Re}\left\{ f(\langle h^*, p(x) \rangle \overline{g(\langle Aw, p(x) \rangle)}) \right\} \right] + \frac{|\alpha_s|}{2} \|Aw\|^{2s} \tag{14}$$

$$= -|\alpha_s| \operatorname{Re}\{\langle Aw, h^* \rangle^s\} + \frac{|\alpha_s|}{2} \|Aw\|^{2s} . \tag{15}$$

We note that Dudeja & Hsu (2018) used a similar trick of a correlational loss containing a single orthogonal polynomial in order to simplify the learning landscape. The global minima of this loss, and in fact the dynamics of gradient flow on it, will be explored in the sequel.

## 3 RELATED WORK

### 3.1 SINGLE INDEX LEARNING

The conditions under which single-index model learning is possible have been well-explored in previous literature. The main assumptions that enable provably learning under gradient flow / gradient descent are monotonicity of the link function (Kakade et al., 2011; Kalai & Sastry, 2009; Shalev-Shwartz et al., 2010; Yehudai & Ohad, 2020) or Gaussian input distribution (Arous et al., 2021). The former assumption essentially corresponds to the setting where the information exponent $s = 1$, as it will have positive correlation with a linear term. Under the latter assumption, the optimal sample complexity was achieved in Damian et al. (2023), with study of learning when the link function is not known in Bietti et al. (2022).

When both assumptions are broken, the conditions on the input distribution of rotation invariance or approximate Gaussianity are nevertheless sufficient for learning guarantees (Bruna et al., 2023). But more unusual distributions, especially in the complex domain that is most convenient for symmetric networks, are not well studied.

### 3.2 SYMMETRIC NEURAL NETWORKS

The primary model for symmetric neural networks was introduced in Zaheer et al. (2017) as the DeepSets model. There are many similar models that enforce permutation invariance (Qi et al., 2017; Santoro et al., 2017; Lee et al., 2019), though we focus on DeepSets because of its relationship with the power sum polynomials and orthogonality (Zweig & Bruna, 2022). We are not aware of any other works that demonstrate provable learning of symmetric functions under gradient-based methods.

## 4 PROVABLY EFFICIENT RECOVERY WITH GRADIENT FLOW

### 4.1 DEFINING THE DYNAMICS

The gradient methods considered in Arous et al. (2021); Ben Arous et al. (2022) are analyzed by reducing to a dimension-free dynamical system of the so-called summary statistics. For instance, in the vanilla single-index model, the summary statistics reduce to the scalar correlation between the learned weight and the true weight. In our case, we have three variables, owing to the fact that the correlation is complex and represented by two scalars, and a third variable controlling the norm of the weight since we aren't using projection.

Note that although our weight vector $w$ is complex, we still apply regular gradient flow to the pair of weight vectors $w_R, w_C$ where $w = w_R + iw_C$. Furthermore, we use the notation $\nabla := \nabla_w = \nabla_{w_R} + i\nabla_{w_C}$. With that in mind, we can summarize the dynamics of our gradient flow in the following Theorem.

**Theorem 4.1.** *Given a parameter $w$, consider the summary statistics $m = \langle Aw, h^* \rangle \in \mathbb{C}$ and $v = \|P_{h^*}^\perp Aw\|^2$ where $P_{h^*}^\perp$ is projection onto the orthogonal complement of $h^*$. Let the polar decomposition of $m$ be $re^{i\theta}$.*

*Then given the preconditioned gradient flow given by*

$$\dot{w} = -\frac{1}{s|\alpha_s|}(A^\dagger A)^{-1}\nabla L(w) , \tag{16}$$

*the summary statistics obey the following system of ordinary differential equations:*

$$\dot{r} = (1 - \delta)r^{s-1}\cos s\theta - (v + r^2)^{s-1}r , \tag{17}$$

$$\frac{d}{dt}\cos s\theta = (1 - \delta)sr^{s-2}(1 - \cos^2 s\theta) , \tag{18}$$

$$\dot{v} = 2\delta r^s \cos s\theta - 2(v + r^2)^{s-1}v , \tag{19}$$

*where $\delta := 1 - \|P_A h^*\|^2$ and $P_A$ is the projection onto the range of $A$.*

The proof is in Appendix D. The main technical details come from using Wirtinger calculus to determine how the real and imaginary parts of $w$ evolve under the flow. Additionally, the correct preconditioner (intuitive from the linear transform of $w$) is crucial for reducing the dynamics to only three summary statistics, and converting to dynamics on $\cos s\theta$ rather than $\theta$ itself simplifies the description of the learning in the next section dramatically.

## 4.2 PROVABLE LEARNING

These dynamics naturally motivate the question of learning efficiency, measured in convergence rates in time in the case of gradient flow. Our main result is that, under some assumptions on the initialization of the frozen weights $\{a_m\}_{m=1}^{M}$ and the initialized weight vector $w_0$, the efficiency is controlled by the initial correlation with the true direction and the information exponent, just as in the Gaussian case.

**Theorem 4.2.** *Consider a fixed $\epsilon > 0$. Suppose the initialization of $w_0$ and $(a_m)_{m=1}^{M}$ are such that:*

(i) *Small correlation and anti-concentration at initialization:* $0 < r_0 \leq 1$,

(ii) *Initial phase condition:* $\cos s\theta_0 \geq 1/2$,

(iii) *Initial magnitude condition for $Aw$:* $\|Aw_0\| = 1$, *or equivalently* $v_0 = 1 - r_0^2$,

(iv) *Small Approximation of optimal error:* $\delta \leq \min(\epsilon/2, O(s^{-s}r_0^4))$.

*Then if we run the gradient flow given in Theorem 4.1 we have $\epsilon$ accuracy in the sense that:*
$$r_T \geq 1 - \epsilon\,, \ \cos s\theta_T \geq 1 - \epsilon\,, \ v_T \leq \epsilon \tag{20}$$
*after time $T$, where depending on the information exponent $s$:*
$$T \leq \begin{cases} O\left(\log \frac{1}{\epsilon}\right) & s = 1\,, \\ O\left(2^{s^2} r_0^{-4s} + \log \frac{1}{\epsilon}\right) & s > 1\,. \end{cases} \tag{21}$$

**Remark 4.3.** *We note that we only recover $\cos s\theta \approx 1$, rather than a guarantee that $\theta \approx 0$, and so the hidden direction is only determined up to scaling by a sth root of unity. This limitation is may appear to be an issue with the choice of the student link function $g$, but it is unavoidable: if the teacher link function $f(z) = \frac{1}{\sqrt{s!}}z^s$, one can calculate that for any choice of $g$, $L(w)$ is invariant to scaling $w$ by an sth root of unity.*

## 4.3 INITIALIZATION GUARANTEES

In order to apply the gradient flow bound proved in Theorem 4.2, it only remains to understand when the assumptions on initialization are met. Unlike the single-index setting with Gaussian inputs, the initial correlation is not guaranteed to be on the scale of $\frac{1}{\sqrt{N}}$, but will depend on the activation function and the random weights in the first layer. Let us introduce the assumptions we'll need:

**Assumption 4.4.** *We assume an analytic activation $\sigma(z) = \sum_{k=0}^{\infty} c_k z^k$, with the notation $\sigma_+ := \max_{1 \leq k \leq N} |c_k|\sqrt{k}$ and $\sigma_- := \min_{1 \leq k \leq \sqrt{N}} |c_k|\sqrt{k}$. We further assume:*

(i) $c_k = 0$ *iff* $k = 0$,

(ii) $\sigma$ *analytic on the unit disk,*

(iii) $1/\sigma_- = O(\text{poly}(N))$,

(iv) $\sum_{k=N+1}^{\infty} k|c_k|^2 \leq e^{-\Omega(\sqrt{N})}$.

The first two conditions are simply required for the application of Proposition 2.3, as the powersum vector $p$ is built out of polynomials induced by the activation and does not include a constant term. The second two conditions concern the decay of the coefficients of $\sigma$, in the sense that the decay must start slow but eventually become very rapid. These conditions are necessary mainly for ensuring the Small Approximation of optimal error condition:

**Lemma 4.5.** *Let $\sigma$ satisfy Assumption 4.4, and assume $M = O(N^3)$. Consider any unit norm $h^* \in \mathbb{C}^\infty$ that is only supported on the first $\sqrt{N}$ elements. If we sample the weights i.i.d. uniformly on the complex unit circle, $a_m \sim S^1$, with probability $1 - 2\exp(-\Omega(N))$:*
$$1 - \|P_A h^*\|^2 \leq e^{-\Omega(\sqrt{N})}\,.$$

Lastly, we can choose an initialization scheme for $w$ which handily ensures the remaining assumptions we need to apply Theorem 4.2. The crucial features of $\sigma$ are similar to the previous result. Namely, we want the initial correlation $r_0$ to be non-negligible because this directly controls the runtime of gradient flow. Slow initial decay with fast late decay of the $\sigma$ coefficients directly implies that $Aw_0$ has a lot of mass in the first $\sqrt{N}$ indices and very little mass past the first $N$ indices. These requirements rule out, say, $\exp$ as an analytic activation because the coefficients decay too rapidly.

**Lemma 4.6.** *Suppose $w$ is sampled from a standard complex Gaussian on $M$ variables. It follows that if we set $w_0 = \frac{w}{\|Aw\|}$, and use the summary statistics from Theorem 4.1, then with probability $1/3 - 2\exp(-\Omega(N))$ and any $h^*$ as in Lemma 4.5*

(i) $1 \geq r_0 \geq c\frac{\sigma_-}{\sigma_+\sqrt{M}}$ *for some universal constant $c > 0$,*

(ii) $\cos s\theta_0 \geq 1/2$,

(iii) $v_0 = 1 - r_0^2$.

Finally, we consider a straightforward choice of $\sigma$ that meets Assumption 4.4 so that we can arrive at an explicit complexity bound on learning:

**Corollary 4.7** (Non-asymptotic Rates for Gradient Flow). *Consider $\xi = 1 - \frac{1}{\sqrt{N}}$ and the specific choice of activation*

$$\sigma(z) = \arctan \xi z + \xi z \arctan \xi z \ .$$

*Suppose we initialize $w$ from a standard complex Gaussian in dimension $M$ with $M = O(N^3)$, and $\{a_m\}_{m=1}^M \sim S^1$ iid. Furthermore, treat $s$ and $\epsilon$ as constants relative to $N$. Then with probability $1/3 - 2\exp(-\Omega(N))$, we will recover $\epsilon$ accuracy in time*

$$T \leq \begin{cases} O\left(\log\frac{1}{\epsilon}\right) & s = 1 \\ O\left(2^{s^2}N^{7s} + \log\frac{1}{\epsilon}\right) & s > 1 \ . \end{cases} \tag{22}$$

*Proof.* By Proposition H.5, the activation $\sigma$ given in the corollary statement satisfies Assumption 4.4, so we can apply Lemma 4.6 and Lemma 4.5 to satisfy the requirements of Theorem 4.2. In particular, the fourth condition is given by assuming $e^{-\Omega(\sqrt{N})} \leq \min(\epsilon/2, O(s^{-s}r_0^4))$ which is true when $s$ is constant, and $\epsilon$ and $r_0$ are at most polynomial compared to $N$.

Note that $\sigma_+ = O(1)$ and $\sigma_- = \Omega\left(\frac{1}{N^{1/4}}\right)$, so it follows that $r_0 = \Omega\left(\frac{1}{N^{7/4}}\right)$ with probability $1/3 - 2\exp(-\Omega(N))$. Conditioning on this bound gives the desired bound on the time for $\epsilon$ accuracy. $\square$

Hence, we have a rate that, for $s = O(1)$, is not cursed by dimensionality to recover the true hidden direction $h^*$. As mentioned above, there are two caveats to this recovery: $w$ is only recovered up to an $s$th root of unity, and to directly make predictions of the teacher model would require using the teacher link function rather than using the student model directly.

Since this result concerns gradient flow over the population loss, a natural question is what barriers exist that stymie the SGD analysis of recent single index papers (Arous et al., 2021; Damian et al., 2023; Bruna et al., 2023). These works treat the convergence of SGD by a standard drift and martingale argument, where the drift follows the population gradient flow, and the martingales are shown to be controlled via standard concentration inequalities and careful arguments around stopping times. Applying these tactics to a discretized version of the dynamics given in Theorem 4.1 mainly runs into an issue during the first phase of training. Unlike in Arous et al. (2021) where the drift dynamics have the correlation monotonically increasing towards 1, at the start of our dynamics the correlation magnitude $r$ and the "orthogonal" part of the learned parameter $v$ are both decreasing (with high probability over the initialization). Showing that this behavior doesn't draw the model towards the saddle point where $r = 0$ requires showing that $v$ decreases meaningfully faster than $r$, i.e. showing that $\frac{d}{dt}\log\frac{r^2}{v}$ is positive. It's not clear what quality of bounds the martingale concentration inequalities would provide for this quantity, and we leave for future work if the six stage proof of the dynamics behavior could be successfully discretized.

## 5 EXPERIMENTS

To study an experimental setup for our setting, we consider the student-teacher setup outlined above with gradient descent. We consider $N = 25$, $M = 100$, and approximate the matrix $A$ by capping the infinite number of rows at 150, which was sufficient for $1 - \|P_A h^*\|^2 \leq 0.001$ in numerical experiments. For the link function $f$, we choose its only non-zero monomial coefficients to be $\alpha_3 = \alpha_4 = \alpha_5 = \frac{1}{\sqrt{3}}$. And correspondingly, $g$ simply has $\alpha_3 = 1$ and all other coefficients at zero.

We choose for convenience an activation function such that $A_{km} = \left(\frac{N-1}{N}\right)^k a_m^k$. We make this choice because, while obeying all the assumptions required in Assumption 4.4, this choice implies that the action of $A$ on the elementary

basis vectors $e_j$ for $1 \leq j \leq \sqrt{N}$ is approximately distributed the same. This choice means that $\|P_A h^*\|$ is less dependent on the choice of $h^*$, and therefore reduces the variance in our experiments when we choose $h^*$ uniformly among unit norm vectors with support on the first $\sqrt{N}$ elements, i.e. uniformly from the complex sphere in degree $\sqrt{N}$.

Under this setup, we train full gradient descent on 50000 samples from the Vandermonde $V$ distribution under 20000 iterations. The only parameter to be tuned is the learning rate, and we observe over the small grid of $[0.001, 0.0025, 0.005]$ that a learning rate of $0.0025$ performs best for the both models in terms of probability of $r$ reaching approximately 1, i.e. strong recovery.

As described in Theorem 4.1, we use preconditioned gradient descent using $(A^\dagger A)^{-1}$ as the preconditioner, which can be calculated once at the beginning of the algorithm and is an easy alteration to vanilla gradient descent to implement. We use the pseudoinverse for improved stability in calculating this matrix, although we note that this preconditioner doesn't introduce stability issues into the updates of our summary statistics, even in the case of gradient descent. Indeed, even if one considers the loss $L(w)$ under an empirical expectation rather than full expectation, the gradient $\nabla L(w)$ can still be seen to be written in the form $A\dagger v$ for some vector $v$. If one preconditions this gradient by $(A^\dagger A)^{-1}$, and observes that the summary statistics $m$ and $v$ both depend on $Aw$ rather than $w$ directly, it follows that the gradient update on these statistics is always of the form $A(A^\dagger A)^{-1}A^\dagger = P_A$, so even in the empirical case this preconditioner doesn't introduce exploding gradients.

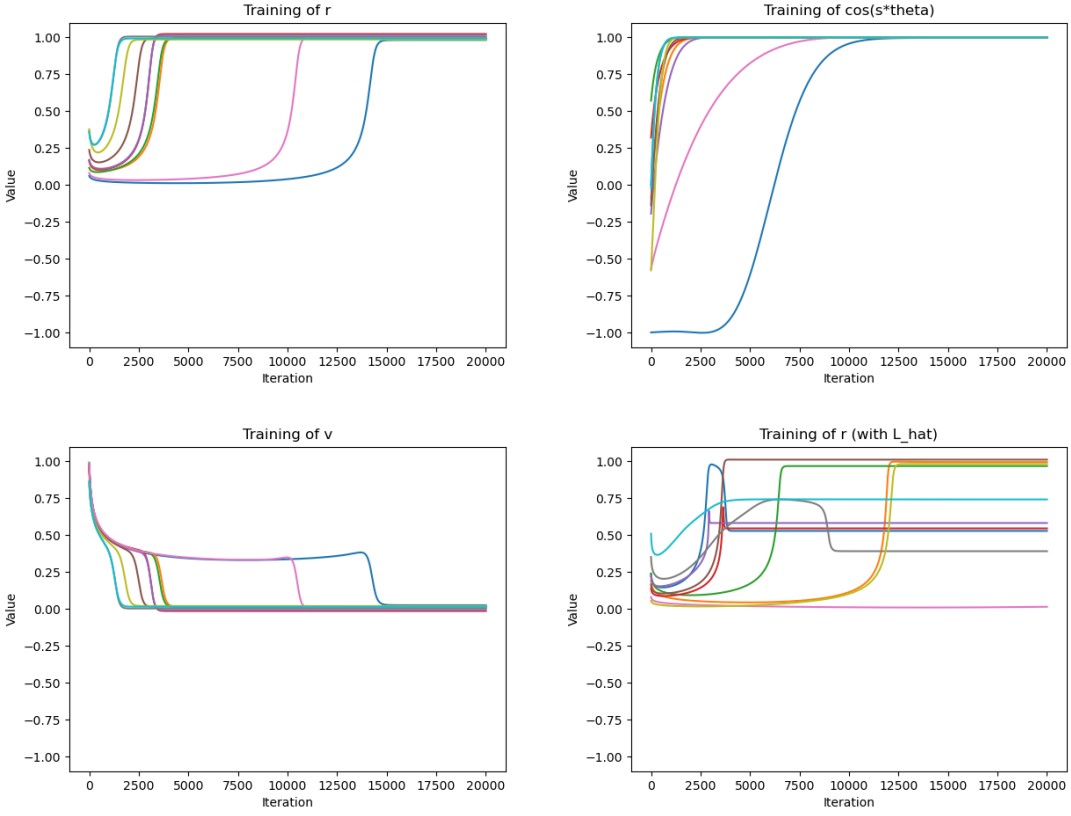

Figure 1: The learning trajectory, over ten independent runs, of the three summary statistics in the case of our chosen loss function $L$, and the trajectory of the $r$ statistic for the more complicated loss function $\hat{L}$

# 6 DISCUSSION

## 6.1 EXPERIMENTAL RESULTS

The outcomes of our experiments are given in Figure 1. We observe very high rates of strong recovery using the loss $L$. For the loss $\hat{L}$, we note that $r$ often becomes stuck, indicating the model has reached a local minima.

We note that our analysis is somewhat pessimistic, as the experimental gradient descent on $L(w)$ will often achieve near perfect accuracy even if $\cos s\theta_0 < 0$. This is mainly an issue of proof technique: although $\cos s\theta$ is always increasing under the dynamics, $r$ is necessarily decreasing for as long as $\cos s\theta$ is negative. It is quite subtle to control whether $\cos s\theta$ will become positive before $r$ becomes extremely small, and the initialization of $r$ is the main feature that controls the runtime of the model. However the empirical results suggest that a chance of success $> 1/2$ is possible under a more delicate analysis.

However, the analysis given in the proof of Theorem 4.2 does accurately capture the brief dip in the value of $r$ in the initial part of training, when the regularization contributes more to the gradient than the correlation until $\cos s\theta$ becomes positive.

Because we can only run experiments on gradient descent rather than gradient flow, we observe the phenomenon of search vs descent studied in Arous et al. (2021), where the increase in the correlation term $r$ is very slow and then abruptly increases. For the model trained with $\hat{L}$, we observe that there is much greater likelihood of failure in the recovery, as $r$ appears to become stuck below the optimal value of 1.

## 6.2 EXTENSIONS

The success of this method of analysis depends heavily on the Hermite-like identity in Proposition 2.3. In general, many of the existing results analyzing single index models need to assume either Gaussian inputs, or uniformly distributed inputs on the Boolean hypercube (see for example Abbe et al. (2023)). In some sense, this works cements the inclusion of the Vandermonde distribution in this set of measures that enable clean analysis. The proof techniques for these three measures are quite disparate, so it remains open to determine if there is a wider class of "nice" distributions where gradient dynamics can be successfully analyzed.

Additionally, the success of the multi-layer training in Bietti et al. (2022); Mahankali et al. (2023) suggests that simultaneously training the frozen first layer weights may not prohibit the convergence analysis. The matrix $A$ depends on the first layer weights through a Vandermonde matrix (see $X$ in the proof of Lemma 4.5), and the simple characterization of the derivative of a Vandermonde matrix alludes to further possibilities for clean analysis.

## 6.3 LIMITATIONS

A first limitation is the focus of this work on complex inputs, analytic activations, and fixed input distribution (namely the squared Vandermonde density). Although complex analytic functions are less commonly studied in the literature, they do still appear in settings like quantum chemistry (Beau & del Campo, 2021; Langmann, 2005). Regarding the focus on the Vandermonde distribution, we note this is similar to the vanilla single-index setting in the restriction to Gaussian inputs, under which the theory is particularly powerful, simplest and understanding of non-Gaussian data is still nascent.

A second limitation is that this work focuses on input distributions over sets of scalars, whereas typically symmetric neural networks are applied to sets of high-dimensional vectors. Proposition 2.3 does not work out of the box for these settings without a high-dimensional analogue of the inner product $\langle \cdot, \cdot \rangle_V$ with similar orthogonality properties. It is possible to define such an inner products on the so-called multisymmetric powersums with similar orthogonality (Zweig & Bruna, 2022), and we leave to future work the question of whether such inner products could grant similar guarantees about the learning dynamics in this more realistic setting.

## 7 CONCLUSION

In this work we've shown a first positive result that quantifies the ability of gradient descent to perform symmetric feature learning, by adapting and extending the tools of two-layer single index models. In essence, this is made possible by a 'miracle', namely the fact that certain powersum expansions under the Vandermonde measure enjoy the same semigroup structure as Hermite polynomials under the Gaussian measure (Proposition 2.3) — leading to a dimension-free summary statistic representation of the loss. Although the resulting dynamics are more intricate than in the Euclidean setting, we are nonetheless able to establish quantitative convergence rates to 'escape the mediocrity' of initialization, recovering the same main ingredients as in previous works Arous et al. (2021); Abbe et al. (2022), driven by the information exponent. To our knowledge, this is the first work to show how learning with gradient based methods necessarily succeeds in this fully non-linear (i.e. not in the NTK regime) setting. Nevertheless, there are many lingering questions.

As discussed, one limitation of the analysis is the reliance on gradient flow rather than gradient descent. We hope that in future work we'll be able to effectively discretize the dynamics, made more challenging by the fact that one must track three parameters rather than simply the correlation. Still, we observe theoretically and empirically that the symmetric single index setting demands a number of unusual choices, such as a correlation loss and distinct student and teacher link function, in order to enable efficient learning. And in a broader scheme, if one remembers the perspective of DeepSets as a very limited form of a three-layer architecture, the issue of provable learning for deeper, more realistic architectures stands as a very important and unexplored research direction — and where Transformers with planted low-dimensional structures appear as the next natural question.

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
