## A  PROOF OF PROPOSITION 2.2

The proposition is true for any activation with a Laurent series, but we will only prove it for activations satisfying Assumption 4.4 since that's the only setting we'll require it.

Consider an analytic activation $\sigma$ with no constant term, given as

$$\sigma(z) = \sum_{k=1}^{\infty} c_k z^k \tag{23}$$

And remind the deepsets features map

$$\Phi_m(x) = \sum_{n=1}^{N} \sigma(a_m x_n) \tag{24}$$

$$\tag{25}$$

where we have neurons without bias terms.

Then for a weight $w \in \mathbb{C}^M$, one can quickly see

$$\langle w, \Phi(x) \rangle = \sum_{m=1}^{M} w_m \sum_{n=1}^{N} \sigma(a_m x_n) \tag{26}$$

$$= \sum_{m=1}^{M} w_m \sum_{n=1}^{N} \sum_{k=1}^{\infty} c_k (a_m x_n)^k \tag{27}$$

$$= \sum_{k=1}^{\infty} \sum_{m=1}^{M} w_m c_k a_m^k \sqrt{k} p_k(x) \tag{28}$$

$$= \langle Aw, p(x) \rangle \tag{29}$$

where $A_{km} = c_k \sqrt{k} a_m^k$

## B  PROOF OF PROPOSITION 2.3

We require some definitions to use the machinery of symmetric polynomials.

**Definition B.1.** *An* integer partition $\lambda$ *is non-increasing, finite sequence of positive integers* $\lambda_1 \geq \lambda_2 \geq \cdots \geq \lambda_k$. *The weight of the partition is given by* $|\lambda| = \sum_{i=1}^{k} \lambda_i$. *The length of a partition* $l(\lambda)$ *is the number of terms in the sequence.*

Then we characterize a product of powersums by:

$$p_\lambda(x) = \prod_i p_{\lambda_i}(x) \tag{30}$$

Finally, define the combinatorial constant $t_\lambda = \prod_{i=1}^{|\lambda|} (m_i)!$ where $m_i$ denotes the number of parts of $\lambda$ equal to $i$.

**Theorem B.2** ((Macdonald, 1998, Chapter VI (9.10)) )**.** *For partitions* $\lambda, \mu$ *with* $|\lambda| \leq N$:

$$\langle p_\lambda, p_\mu \rangle_V = t_\lambda \mathbb{1}_{\lambda=\mu} \tag{31}$$

With that in mind, let's consider the inner product of two simple single-index functions.

Let $p = [p_1, p_2, \dots]$ be an infinite vector of powersums, and choose exponents $i, j$ with $i \leq \sqrt{N}$. Then for any $h, \tilde{h} \in \mathbb{C}^\infty$ such that $h$ is only supported on the first $\sqrt{N}$ entries:

$$\langle \langle h, p \rangle^i, \langle \tilde{h}, p \rangle^j \rangle_V = \left\langle \sum_{|\alpha|=i} \binom{i}{\alpha} h^\alpha \overline{p^\alpha}, \sum_{|\alpha|=j} \binom{j}{\alpha} \tilde{h}^\alpha \overline{p^\alpha}, \right\rangle_V \tag{32}$$

$$= \delta_{ij} \sum_{|\alpha|=i} \binom{i}{\alpha}^2 \langle p^\alpha, p^\alpha \rangle_V h^\alpha \overline{\tilde{h}^\alpha} \tag{33}$$

$$= \delta_{ij} \sum_{|\alpha|=i} \binom{i}{\alpha}^2 \left( \prod_{k=1}^{\sqrt{N}} \alpha_k! \right) h^\alpha \overline{\tilde{h}^\alpha} \tag{34}$$

$$= \delta_{ij} i! \sum_{|\alpha|=i} \binom{i}{\alpha} h^\alpha \overline{\tilde{h}^\alpha} \tag{35}$$

$$= \delta_{ij} i! \langle h, \tilde{h} \rangle^i \tag{36}$$

## C    PROOF OF PROPSITION 2.6

Applying Proposition 2.3 and using assumptions on the degree bound on $f$ and the support of $h^*$, we can write:

$$\langle f(\langle h^*, p \rangle), f(\langle Aw, p \rangle) \rangle_V = \left\langle \sum_{j=1}^{\sqrt{N}} \frac{\alpha_j}{\sqrt{j!}} \langle h^*, p \rangle^j, \sum_{j=1}^{\sqrt{N}} \frac{\alpha_j}{\sqrt{j!}} \langle Aw, p \rangle^j \right\rangle_V \tag{37}$$

$$= \sum_{j=1}^{\sqrt{N}} |\alpha_j|^2 \langle h^*, Aw \rangle^j \tag{38}$$

Hence we have

$$\hat{L}(w) = E_{x \sim V} \left[ -\operatorname{Re}\left\{ f(\langle h^*, p(x) \rangle \overline{f(\langle Aw, p(x) \rangle)}) \right\} \right] + \sum_{j=1}^{\sqrt{N}} \frac{|\alpha_j|^2}{2} \|Aw\|^{2j} \tag{39}$$

$$= -\frac{1}{2} \langle f(\langle h^*, p \rangle), f(\langle Aw, p \rangle) \rangle_V - \frac{1}{2} \overline{\langle f(\langle h^*, p \rangle), f(\langle Aw, p \rangle) \rangle_V} + \sum_{j=1}^{\sqrt{N}} \frac{|\alpha_j|^2}{2} \|Aw\|^{2j} \tag{40}$$

$$= -\sum_{j=1}^{\sqrt{N}} |\alpha_j|^2 \operatorname{Re}\{\langle h^*, Aw \rangle^j\} + \frac{|\alpha_j|^2}{2} \|Aw\|^{2j} \tag{41}$$

Now, we use the same notation as in Theorem 4.1 and introduce variables $m = \langle Aw, h^* \rangle = re^{i\theta}$ and $v = \|Aw\|^2 - r^2$, such that we can write:

$$\hat{L}(w) = \sum_{j=1}^{\sqrt{N}} |\alpha_j|^2 \left( -r^j \cos j\theta + \frac{1}{2}(v + r^2)^j \right) \tag{42}$$

Because $r \geq 0$ and $v \geq 0$, this loss can be minimized by setting $v = 0$ and any $\theta$ where $\cos j\theta = 1$ for all $j$ with $\alpha_j \neq 0$. Since we assume there are distinct indices $i, j$ that are coprime with non-zero support, we require $i\theta$ and $j\theta$ to both be multiples of $2\pi$, which is only possible if $\theta \equiv 0 \mod 2\pi$. Therefore:

$$\hat{L}(w) = \sum_{j=1}^{\sqrt{N}} |\alpha_j|^2 \left( -r^j + \frac{1}{2} r^{2j} \right) \tag{43}$$

$$= C + \sum_{j=1}^{\sqrt{N}} \frac{|\alpha_j|^2}{2} \left( r^j - 1 \right)^2 \tag{44}$$

for some constant $C$, and this is minimized at $r = 1$. Hence, if $r = 1, \theta \equiv 0 \mod 2\pi, v = 0$, it follows that $Aw = h^*$.

## D PROOF OF THEOREM 4.1

Given the matrix $A$ and weight $w$, an identical calculation to the one in Proposition 2.6 shows that

$$L(w) = E_{x \sim V} \left[ -\operatorname{Re} \left\{ f(\langle h^*, p(x) \rangle \overline{g(\langle Aw, p(x) \rangle)}) \right\} \right] + \frac{|\alpha_s|}{2} \|Aw\|^{2s} \tag{45}$$

$$= -|\alpha_s| \operatorname{Re}\{\langle Aw, h^* \rangle^s\} + \frac{|\alpha_s|}{2} \|Aw\|^{2s} \tag{46}$$

To calculate the gradient with respect to the real and imaginary parts of $w$, we use tools from Wirtinger calculus (Fischer, 2005). Using the notation that $\nabla_{\overline{w}} = \frac{1}{2}(\nabla_{w_R} + i\nabla_{w_C})$ and the appropriate generalization of the chain rule, we have:

$$2\nabla_{\overline{w}} \operatorname{Re}\{\langle Aw, h^* \rangle^s\} = \nabla_{\overline{w}} \left( \langle Aw, h^* \rangle^s + \overline{\langle Aw, h^* \rangle^s} \right) \tag{47}$$

$$= \nabla_{\overline{w}} \overline{\langle Aw, h^* \rangle}^s \tag{48}$$

$$= s\overline{\langle Aw, h^* \rangle}^{s-1} A^\dagger h^* \tag{49}$$

Likewise,

$$2\nabla_{\overline{w}} \|Aw\|^{2s} = 2s\|Aw\|^{2(s-1)} \nabla_{\overline{w}} \|Aw\|^2 \tag{50}$$

$$= 2s\|Aw\|^{2(s-1)} \nabla_{\overline{w}} \left( w^\dagger A^\dagger Aw \right) \tag{51}$$

$$= 2s\|Aw\|^{2(s-1)} A^\dagger Aw \tag{52}$$

Thus, we have:

$$\nabla L = \nabla_{w_R} L + i\nabla_{w_C} L \tag{53}$$

$$= 2\nabla_{\overline{w}} L \tag{54}$$

$$= -s|\alpha_s| \overline{\langle Aw, h^* \rangle}^{s-1} A^\dagger h^* + s|\alpha_s| \|Aw\|^{2(s-1)} A^\dagger Aw \tag{55}$$

We introduce the parameters

$$m = \langle Aw, h^* \rangle = \langle w, A^\dagger h^* \rangle \tag{56}$$

$$v = \|P_{h^*}^\perp Aw\|^2 = \|Aw\|^2 - |m|^2 \tag{57}$$

And we consider preconditioned gradient flow of the form (where for complex variables we use similar notation that $\dot{w} = \dot{w_R} + \dot{w_C} i$):

$$\dot{w} = -\frac{1}{s|\alpha_s|}(A^\dagger A)^{-1}\nabla L \tag{58}$$

$$= \overline{m}^{s-1}(A^\dagger A)^{-1}A^\dagger h^* - \|Aw\|^{2(s-1)}w \tag{59}$$

It follows that

$$\dot{m} = \langle \dot{w}, A^\dagger h^* \rangle \tag{60}$$

$$= \|P_A h^*\|^2 \overline{m}^{s-1} - (v + |m|^2)^{s-1}m \tag{61}$$

where $P_A = A(A^\dagger A)^{-1}A^\dagger$ is the orthogonal projection onto the range of $A$.

Let $m = a + bi = re^{i\theta}$, so we have $\dot{m} = \dot{a} + \dot{b}i$. Thus

$$\dot{a} = \|P_A h^*\|^2 r^{s-1}\cos(s-1)\theta - (v + r^2)^{s-1}r\cos\theta \tag{62}$$

$$\dot{b} = -\|P_A h^*\|^2 r^{s-1}\sin(s-1)\theta - (v + r^2)^{s-1}r\sin\theta \tag{63}$$

Now we do a change of variables, because $a = r\cos\theta$ and $b = r\sin\theta$, so

$$\dot{a} = \dot{r}\cos\theta - r\dot{\theta}\sin\theta \tag{64}$$

$$\dot{b} = \dot{r}\sin\theta + r\dot{\theta}\cos\theta \tag{65}$$

$$\tag{66}$$

Rearranging, we can get the flow on $r$ and $\theta$:

$$\dot{r} = \dot{a}\cos\theta + \dot{b}\sin\theta \tag{67}$$

$$= \|P_A h^*\|^2 r^{s-1}\cos s\theta - (v + r^2)^{s-1}r \tag{68}$$

$$r\dot{\theta} = -\dot{a}\sin\theta + \dot{b}\cos\theta \tag{69}$$

$$= -\|P_A h^*\|^2 r^{s-1}\sin s\theta \tag{70}$$

$$\tag{71}$$

We can instead control the flow on $\cos s\theta$:

$$\frac{d}{dt}\cos s\theta = -\dot{\theta}s\sin s\theta = \|P_A h^*\|^2 s r^{s-2}\sin^2 s\theta \tag{72}$$

and calculate the flow on $v$:

$$\dot{v} = 2\operatorname{Re}\{\langle A\dot{w}, Aw\rangle\} - 2r\dot{r} \tag{73}$$

$$= 2\left(r^s\cos s\theta - (v + r^2)^s - \|P_A h^*\|^2 r^s\cos s\theta + (v + r^2)^{s-1}r^2\right) \tag{74}$$

$$= 2(1 - \|P_A h^*\|^2)r^s\cos s\theta - 2(v + r^2)^{s-1}v \tag{75}$$

Finally, introducing the notation $\delta = 1 - \|P_A h^*\|^2$, we have

$$\dot{r} = (1 - \delta)r^{s-1}\cos s\theta - (v + r^2)^{s-1}r \tag{76}$$

$$\frac{d}{dt}\cos s\theta = (1 - \delta)s r^{s-2}(1 - \cos^2 s\theta) \tag{77}$$

$$\dot{v} = 2\delta r^s\cos s\theta - 2(v + r^2)^{s-1}v \tag{78}$$

# E PROOF OF THEOREM 4.2

We will use the following facts repeatedly in the below arguments.

First, because $\dot{r} \geq 0$ when $r = 0$, and $\dot{r} \leq 0$ when $r = 1$, it follows that $r$ can never leave the range $[0, 1]$. Furthermore, note that $\cos s\theta$ is always non-decreasing.

## E.1 CASE $s = 1$

In the setting with information complexity equal to 1, we immediately have the following identities:

$$\dot{r} = (1 - \delta) \cos \theta - r \tag{79}$$

$$\frac{d}{dt} \cos \theta \geq (1 - \delta)(1 - \cos^2 \theta) \tag{80}$$

$$\dot{v} \leq 2\delta - 2v \tag{81}$$

Let us address $v$ first. From our assumptions, $\delta < \epsilon$, and so when $v \geq \epsilon$, $\dot{v}$ is negative. It follows that a trajectory that begins below $\epsilon$ cannot ever exceed $\epsilon$. In other words, if $v_0 \leq \epsilon$, $v$ can never exceed $\epsilon$ and we've achieved optimality.

Otherwise, supposing $v_0 > \epsilon$, consider values of $t$ where $v_t > \delta$ so that the RHS of the inequality of $\dot{v}$ is strictly negative and we may write:

$$\frac{\dot{v}}{\delta - v} \geq 2 \tag{82}$$

Integrating from $0$ to $t$ gives that

$$-\log |\delta - v_t| - (-\log |\delta - v_0|) \geq 2t \tag{83}$$

which yields the bound

$$v_t \leq \delta + (v_0 - \delta)e^{-2t} \leq \delta + e^{-2t} \tag{84}$$

By Lemma H.1,

$$\cos \theta_t \geq \tanh((1 - \delta)t) \tag{85}$$

Finally, we consider $r$.

Choose $T_1 = \inf\{t \geq 0 : v_t \leq \epsilon, \cos \theta_t \geq \frac{1 - \epsilon/2}{1 - \delta}\}$, and $T_2 = \inf\{t \geq T_1 : r_t \geq 1 - \epsilon\}$. Note that one can easily confirm that $T_1 \leq O\left(\log \frac{1}{\epsilon}\right)$

Then for all $t \in [T_1, T_2)$, we have

$$\dot{r}_t = (1 - \delta) \cos \theta_t - r_t \geq 1 - \epsilon/2 - r_t \tag{86}$$

and the RHS is always non-negative.

Dividing by the RHS and integrating from $T_1$ to $t$ gives

$$-\log(1 - \epsilon/2 - r_t) + \log(1 - \epsilon/2 - r_{T_1}) \geq t - T_1 \tag{87}$$

Rearranging gives

$$r_t \geq 1 - \epsilon/2 - (1 - \epsilon/2 - r_{T_1})e^{T_1 - t} \tag{88}$$

Note that by definition of $T_2$, it follows that

$$1 - r_t \leq \epsilon/2 + e^{T_1 - t} \tag{89}$$

So it follows that $T_2 \leq T_1 + \log \frac{2}{\epsilon}$.

Altogether, the total time to achieve $\epsilon$ optimality for all three variables is $O\left(\log \frac{1}{\epsilon}\right)$.

### E.2 CASE $s > 1$

In this case, because we cannot straightforwardly solve or bound the system of ODEs, we need to control rates in stages. We have a stopping time for one variable at a time, and use local monotonicity to ensure bounds on the remaining variables.

**First Phase** In the first stage, we consider the duration of time $T_1 = \inf\{t \geq 0 : v_t \leq v^*\}$ where $v^* := 2^{-s}6^{-2}s^{-2}r_0^4$, and bound the behavior of each variable. Below, we will consider $t \in [0, T_1]$.

To control the behavior or $r$, we consider the following manipulations:

$$\frac{d}{dt} \log r^2 = 2(1 - \delta)r^{s-2} \cos s\theta - 2(v + r^2)^{s-1} \tag{90}$$

$$\frac{d}{dt} \log v = 2\delta \frac{r^s \cos s\theta}{v} - 2(v + r^2)^{s-1} \tag{91}$$

This implies

$$\frac{d}{dt} \log \frac{r^2}{v} = 2r^{s-2} \cos s\theta \left(1 - \delta - \delta \frac{r^2}{v}\right) \tag{92}$$

By definition, in this range of $t$ we have $v_t > \frac{\delta}{1-\delta}$, so it follows that the RHS of this equation is always positive. Hence it follows that $\log \frac{r^2}{v}$ is increasing, and by monotonicity of $\log$, we have

$$\frac{r^2}{v} \geq \frac{r_0^2}{v_0} \geq r_0^2 \tag{93}$$

This implies that

$$\dot{r} = (1 - \delta)r^{s-1} \cos s\theta - (v + r^2)^{s-1}r \tag{94}$$

$$\geq (1 - \delta)r^{s-1} \cos s\theta - \left(\frac{r^2}{r_0^2} + r^2\right)^{s-1} r \tag{95}$$

$$\geq r^{s-1}\left((1 - \delta)\cos s\theta - \left(\frac{1}{r_0^2} + 1\right)^{s-1} r^s\right) \tag{96}$$

Suppose it is true that $r \leq \frac{1}{6}r_0^2$, then it follows that:

$$r \leq \frac{r_0^2(1-\delta)\cos s\theta_0}{2} \tag{97}$$

$$\leq \frac{r_0^2(1-\delta)\cos s\theta_0}{r_0^2+1} \tag{98}$$

$$= \frac{(1-\delta)\cos s\theta_0}{\frac{1}{r_0^2}+1} \tag{99}$$

$$\leq \frac{((1-\delta)\cos s\theta)^{1/s}}{\left(\frac{1}{r_0^2}+1\right)^{\frac{s-1}{s}}} \tag{100}$$

So it follows that $\dot{r}$ will be positive whenever $r \leq \frac{1}{6}r_0^2$. We have $r_0 \geq \frac{1}{6}r_0^2$, it follows that $r_t \geq \frac{1}{6}r_0^2$ for $t \leq T_1$.

Finally we can control $v$ by observing that, for $t \in [0, T_1]$, $v \geq v^* \geq (2\delta)^{1/s}$. Hence,

$$\dot{v} \leq 2\delta - 2v^s \leq -v^s \tag{101}$$

which implies

$$-\frac{\dot{v}}{v^s} \geq 1 \tag{102}$$

And integrating from 0 to $t \leq T_1$ gives

$$v_t^{-(s-1)} \geq \frac{1}{s-1}v_t^{-(s-1)} - \frac{1}{s-1}v_0^{-(s-1)} \geq t \tag{103}$$

Rearranging gives

$$v_t \leq t^{-\frac{1}{s-1}} \tag{104}$$

This gives a bound on $T_1 \leq (v^*)^{-(s-1)} = O(2^{s^2}r_0^{-4s})$

Lastly by monotonicity we have $\cos s\theta_{T_1} \geq \cos s\theta_0$.

So to summarize:

$$r_{T_1} \geq \frac{1}{6}r_0^2 \tag{105}$$

$$\cos s\theta_{T_1} \geq \cos s\theta_0 \tag{106}$$

$$v_{T_1} \leq v^* \tag{107}$$

Furthermore, we've actually proven that $v_t \leq v^*$ for all $t \geq T_1$, which we will use in subsequent phases.

**Second Phase** We define $T_2 = \inf\{t \geq T_1 : r_t \geq 1/5\}$. As before, if $r_{T_1} \geq 1/5$ then $T_2 = 0$ and we can skip to the next phase, so we assume otherwise.

Using the identity $(1+x)^k \leq 1 + 2^k x$ which holds for any $x \in [0, 1]$ and $k \geq 1$, observe that the ODE governing $r$ can now be bounded as:

$$\dot{r} = (1 - \delta)\cos s\theta r^{s-1} - (v + r^2)^{s-1} r \tag{108}$$

$$\geq (1 - \delta)\cos s\theta_0 r^{s-1} - \left(\frac{v}{r^2} + 1\right)^{s-1} r^{2s-1} \tag{109}$$

$$\geq (1 - \delta)\cos s\theta_0 r^{s-1} - \left(1 + 2^{s-1}\frac{v}{r^2}\right) r^{2s-1} \tag{110}$$

$$\geq \frac{1-\delta}{2} r^{s-1} - \left(1 + \frac{r_0^4}{2s^2(6r)^2}\right) r^{2s-1} \tag{111}$$

where in the last step we use that $v \leq v^*$ and plug in the definition of $v^*$ and the bound $\cos s\theta_0 \geq 1/2$.

Consider any $t$ when $r = \frac{1}{6}r_0^2$, and observe that the above inequality implies $\dot{r} > 0$. Because $r_{T_1} \geq \frac{1}{6}r_0^2$, this implies we will always have $r \geq \frac{1}{6}r_0^2$ for larger values of $t$, and we may bound:

$$\dot{r} \geq \frac{1-\delta}{2} r^{s-1} - \left(1 + \frac{1}{2s^2}\right) r^{2s-1} \tag{112}$$

Hence, we can apply Lemma H.2 with $a = (1 - \delta)/2$, $b = 1 + \frac{1}{2s^2}$, where $k^2 = (a/b)^2 \geq 1/5$, and using the initialization of $r_{T_1}$. This grants the bound that $T_2 \leq T_1 + O(s^4 r_{T_1}^{-s+1}) = T_1 + O(6^s r_0^{-2s+2})$.

Therefore the new summary is:

$$r_{T_2} \geq 1/5 \tag{113}$$
$$\cos s\theta_{T_2} \geq \cos s\theta_0 \tag{114}$$
$$v_{T_2} \leq v^* \tag{115}$$

**Third Phase**   We define $T_3 = \inf\{t \geq T_2 : \cos s\theta_t \geq \frac{1 - \frac{1}{4s^4}}{1-\delta}\}$

First of all, note that the bound on $r$ derived in the last phase required lower bounding $\cos s\theta$ by $\cos s\theta_0$. Since $\cos s\theta$ is non-decreasing, that bound is still true by an identical argument.

So we can bound the ODE for $\theta$:

$$\frac{d}{dt}\cos s\theta = (1 - \delta)s r^{s-2}(1 - \cos^2 s\theta) \tag{116}$$

$$\geq (1 - \delta)s(1/5)^{s-2}(1 - \cos^2 s\theta) \tag{117}$$

Note that by lemma H.1 with $k = (1 - \delta)s(1/5)^{s-2}$, we have

$$T_3 \leq T_2 + O(5^s \log s) \tag{118}$$

The bound $v \leq v^*$ continues to hold. In summary, we now have:

$$r_{T_3} \geq 1/5 \tag{119}$$

$$\cos s\theta_{T_3} \geq \frac{1 - \frac{1}{4s^4}}{1 - \delta} \tag{120}$$

$$v_{T_3} \leq v^* \tag{121}$$

**Fourth Phase**   We define $T_4 = \inf\{t \geq T_3 : r_t \geq r^*\}$ where $r^* := 1 - \frac{1}{s^2}$. Again, consider the non-trivial case where $T_4 \neq 0$.

Because the bound on $v$ is the same, and the bound on $\cos s\theta$ is better than before, we can now bound the ODE of $r$ similarly to the second phase:

$$\dot{r} \geq \left(1 - \frac{1}{4s^4}\right) r^{s-1} - \left(1 + \frac{1}{2s^2}\right) r^{2s-1} \tag{122}$$

Applying Lemma H.2 with $k = \frac{1 - \frac{1}{4s^4}}{1 + \frac{1}{2s^2}} = 1 - \frac{1}{2s^2}$, we have:

$$T = \inf\{t \geq T_3 : r \geq k^2\} \leq T_3 + O(5^s \log s) \tag{123}$$

Finally, note that $k^2 = \left(1 - \frac{1}{2s^2}\right)^2 \geq 1 - \frac{1}{s^2}$, which implies that $T_4 \leq T$.

Thus we have:

$$r_{T_4} \geq r^* \tag{124}$$

$$\cos s\theta_{T_4} \geq \frac{1 - \frac{1}{4s^4}}{1 - \delta} \tag{125}$$

$$v_{T_4} \leq v^* \tag{126}$$

**Fifth Phase**   We define $T_5 = \inf\{t \geq T_4 : \cos s\theta_t \geq \frac{1 - \epsilon/2}{1 - \delta}, v_t \leq v^\dagger\}$ where $v^\dagger = 2^{-s}(\epsilon/2)(r^*)^2$.

Again, since $\cos s\theta$ is increasing and $v$ is always less than $v^*$, the bound on $r \geq r^*$ established in the last step will stay true.

Thus, by the identity $r^k \geq (r^*)^k = \left(1 - \frac{1}{s^2}\right)^k \geq 1 - \frac{k}{s^2}$ we have the ODE inequalities:

$$\frac{d}{dt} \cos s\theta = (1 - \delta)sr^{s-2}(1 - \cos^2 s\theta) \tag{127}$$

$$\geq (1 - \delta)s\left(1 - \frac{1}{s}\right)(1 - \cos^2 s\theta) \tag{128}$$

$$\dot{v} = 2\delta r^s \cos s\theta - 2(v + r^2)^{s-1}v \tag{129}$$

$$\leq 2\delta - 2\left(1 - \frac{2(s-1)}{s^2}\right)v \tag{130}$$

It is easy to see that we'll have the bound

$$T_5 \leq T_4 + O\left(\log \frac{1}{\epsilon}\right) \tag{131}$$

and in summary

$$r_{T_5} \geq r^* \tag{132}$$

$$\cos s\theta_{T_5} \geq \frac{1 - \epsilon/2}{1 - \delta} \tag{133}$$

$$v_{T_5} \leq v^\dagger \tag{134}$$

**Sixth Phase**   We define $T_6 = \inf\{t \geq T_5 : r_t \geq 1 - \epsilon\}$, and assume the non-trivial setting where $T_6 \neq 0$.

Note that $\dot{v}$ is negative when $v = v^\dagger$, so the bound $v \leq v^\dagger$ remains true for $t \geq T_5$. Thus, we can control the ODE of $r$ one more time:

$$\dot{r} = (1 - \delta)r^{s-1}\cos s\theta - (v + r^2)^{s-1}r \tag{135}$$

$$\geq (1 - \delta)r^{s-1}(1 - \epsilon/2) - \left(1 + \frac{v}{r^2}\right)^{s-1} r \tag{136}$$

$$\geq (1 - \epsilon/2)r^{s-1} - \left(1 + 2^s\frac{v^\dagger}{r^2}\right) r^{2s-1} \tag{137}$$

$$\geq (1 - \epsilon/2)r^{s-1} - \left(1 + \epsilon/2\frac{(r^*)^2}{r^2}\right) r^{2s-1} \tag{138}$$

One can confirm that when $r = r^*$, the RHS of the above inequality is positive, so $\dot{r} \geq 0$. Thus, since $r_{T_5} \geq r^*$, it will always be the case that $r \geq r^*$ for $t \geq T_5$, so as before we bound:

$$\dot{r} \geq (1 - \epsilon/2)r^{s-1} - (1 + \epsilon/2)r^{2s-1} \tag{139}$$

By Lemma H.2, we have that

$$T_6 \leq T_5 + O\left(\log\frac{1}{\epsilon}\right) \tag{140}$$

and thus we've achieved $\epsilon$ optimality for all three of our variables.

## F   PROOF OF LEMMA 4.5

Remind from Proposition 2.2 that $A \in \mathbb{C}^{\infty \times M}$ is of the form

$$A_{km} = c_k\sqrt{k}a_m^k \tag{141}$$

where we assume $c_k > 0$, and $a_m \sim S^1$. Note that

$$1 - \|P_A h^*\|^2 = \|P_A^\perp h^*\|^2 \tag{142}$$

$$= \min_w \|Aw - h^*\|^2 \tag{143}$$

$$\tag{144}$$

so we need to choose a candidate value of $w$.

Consider the block decomposition

$$A = \left[\frac{B}{C}\right] \tag{145}$$

where $B \in \mathbb{C}^{N \times M}$ and $C \in \mathbb{C}^{\infty \times M}$. Suppose we decompose $h^* = \left[\frac{u}{0}\right]$ where $u \in \mathbb{C}^N$. Then if we apply the pseudoinverse and define $w = B^+u$, observe:

$$Aw = \left[\frac{B}{C}\right] B^+u \tag{146}$$

$$= \left[\frac{BB^+u}{CB^+u}\right] \tag{147}$$

Observe that we can decompose $B = DX$ where $D$ is a diagonal matrix such that $D_{kk} = c_k\sqrt{k}$ and $X_{km} = a_m^k$. Since $N < M$, one can see $X$ is a rectangular Vandermonde matrix evaluated on $\{a_m\}_{m=1}^M$. Almost surely, these values are all pairwise distinct, which implies that $X$ has linearly independent rows. Since $D$ is diagonal with no zeros along the diagonal, $B$ also has linearly independent rows. This condition implies $BB^+ = I$. So we have

$$Aw = \left[\frac{u}{CB^+u}\right] \tag{148}$$

Remember $\|u\| = \|h^*\| = 1$, as $u$ is the first $N$ elements of $h^*$ and hence still only supported on the first $\sqrt{N}$ elements. Because $B^+ = X^+D^{-1}$, we have:

$$\|CB^+u\| \leq \|C\|\|X^+\|\|D^{-1}u\| \tag{149}$$
$$\tag{150}$$

We can now go about bounding these norms.

Since $u$ is only supported on the first $\sqrt{N}$ elements and $\|u\| = 1$, it follows $\|D^{-1}u\| \leq \max_{1 \leq k \leq \sqrt{N}}\left|\frac{1}{c_k\sqrt{k}}\right| = \frac{1}{\sigma_-}$.

By Lemma H.4, we have the bound

$$\|X^+\| \leq O\left(\frac{1}{\sqrt{M}}\right) \tag{151}$$

Finally for any $\hat{w} \in \mathbb{C}^M$ with $\|\hat{w}\| = 1$, we have by Cauchy-Schwarz:

$$\|Cw\|^2 = \sum_{k=N+1}^{\infty}\left|\sum_{m=1}^{M}\hat{w}_m c_k\sqrt{k}a_m^k\right|^2 \tag{152}$$
$$\leq \sum_{k=N+1}^{\infty}\|\hat{w}\|^2\sum_{m=1}^{M}\left|c_k\sqrt{k}\right|^2 \tag{153}$$
$$= M\sum_{k=N+1}^{\infty}k|c_k|^2 \tag{154}$$
$$\leq Me^{-\Omega(\sqrt{N})} \tag{155}$$

where we use in the last step Assumption 4.4.

With these bounds, we clearly have

$$1 - \|P_A h^*\| \leq \|Aw - h^*\|^2 \tag{156}$$
$$= \left\|\left[\frac{u}{CB^+u}\right] - \left[\frac{u}{0}\right]\right\|^2 \tag{157}$$
$$\leq \|CB^+u\|^2 \tag{158}$$
$$\leq \frac{M}{\sqrt{M}\sigma_-}e^{-\Omega(\sqrt{N})} \tag{159}$$

Because $M = O(N^3)$, and we've assumed $1/\sigma_-$ is polynomial in $N$, this bound can be written as $e^{-\Omega(\sqrt{N})}$ for possibly different constants in the big $O$ notation.

## G    PROOF OF LEMMA 4.6

Remind that $m_0 = \langle Aw_0, h^* \rangle = \frac{1}{\|Aw\|}\langle Aw, h^* \rangle$. Because the complex Gaussian is invariant to multiplication by an unit modulus complex number, it follows that $\theta_0$ is independent of $r_0$ and uniformly distributed on $S^1$. Because $s$ is a positive integer, $s\theta_0$ is also uniformly distributed on $S^1$, and hence $P(\cos s\theta_0 \geq 1/2) = 1/3$. And by our choice of normalization, $v_0 = 1 - r_0^2$ automatically. So it only remains to prove the first statement is true with high probability.

We remind that $r_0 = \frac{|\langle Aw, h^* \rangle|}{\|Aw\|}$. By Cauchy-Schwartz, it's clear that $r_0 \leq 1$, so only the lower bound is non-trivial. If we use the same notation to decompose the matrix $A$ as in the proof of Lemma 4.5, it's clear that

$$|\langle Aw, h^* \rangle| = |\langle Bw, u \rangle| \tag{160}$$
$$= |\langle w, B^\dagger u \rangle| \tag{161}$$

If we condition on $B$, then by rotation invariance of the Gaussian, note that $|\langle w, B^\dagger u \rangle|$ is distributed identically to $|g|\|B^\dagger u\|$ where $g$ is sampled from a one dimensional complex Gaussian.

By the argument in Lemma 4.6, since $u$ is only supported on the first $\sqrt{N}$ elements, note that:

$$\|B^\dagger u\| = \|X^\dagger D^\dagger u\| \tag{162}$$
$$\geq \sigma_N(X)\|D^\dagger u\| \tag{163}$$
$$\geq \sigma_N(X)\sigma_- \tag{164}$$
$$\geq \sigma_- O(\sqrt{M}) \tag{165}$$

with probability $1 - 2\exp(-\Omega(N))$ by Lemma H.4

Lastly, we need to control

$$\|Aw\| \leq \|Bw\| + \|Cw\| \leq (\|B\| + \|C\|)\|w\| \tag{166}$$

And we can write again by Lemma H.4, with similarly high probability:

$$\|B\| = \|DX\| \tag{167}$$
$$\leq \|D\|\|X\| \tag{168}$$
$$\leq \sigma_+ \sigma_1(X) \tag{169}$$
$$\leq \sigma_+ O(\sqrt{M}) \tag{170}$$

Combining this with the bound on $\|C\|$ we derived in Lemma 4.6, and the concentration on $\|w\|$ from Lemma H.3 we have with probability $1 - 2\exp(-\Omega(N))$:

$$\|Aw\| \leq \left(\sigma_+ O(\sqrt{M}) + e^{-\Omega(\sqrt{N})}\right) O(\sqrt{M}) \tag{171}$$

Finally we can say that with probability $1 - 2\exp(-\Omega(N))$

$$r_0 \geq c\frac{\sigma_-}{\sigma_+\sqrt{M}} \tag{172}$$

for some universal constant $c$.

# H    AUXILIARY LEMMAS

## H.1    DYNAMICS INEQUALITY LEMMAS

The following lemmas provide bounds on our dynamics that we can apply multiple times in different phases of the proof. Both of these lemmas are essentially special cases of the Bihari-LaSalle Inequality (Bihari, 1956), but because the proofs are much simplified due to our setting, and for completeness, we include the proofs below.

**Lemma H.1.** *Consider $\theta$ with the differential inequality*

$$\frac{d}{dt} \cos s\theta \geq k(1 - \cos^2 s\theta) \tag{173}$$

*with $\cos s\theta_0 \geq 1/2$. Then we have*

$$\cos s\theta_t \geq \tanh(kt) \tag{174}$$

*and hence if $T = \inf\{t \geq 0 : \cos s\theta_t \geq c\}$, then $T \leq \frac{1}{2k} \log \frac{2}{1-c}$.*

*Proof.* Clearly the RHS of the inequality is always positive, so we may write:

$$\frac{\frac{d}{dt} \cos s\theta}{1 - \cos^2 s\theta} \geq k \tag{175}$$

and integrating from $0$ to $t$ gives

$$\tanh^{-1}(\cos s\theta_t) - \tanh^{-1}(\cos s\theta_0) \geq kt \tag{176}$$

Note $\tanh^{-1}(\cos s\theta_0) \geq 0$, so $\cos s\theta_t \geq \tanh(kt)$. Since $\cos s\theta_t$ is increasing, it follows that

$$T \leq \frac{\tanh^{-1}(c)}{k} \tag{177}$$

And using the closed form of $\tanh^{-1}(c)$ for $|c| < 1$ implies

$$T \leq \frac{1}{2k} \log \frac{1+c}{1-c} \tag{178}$$

$$\leq \frac{1}{2k} \log \frac{2}{1-c} \tag{179}$$

$\square$

**Lemma H.2.** *Consider $s \geq 2$. Suppose we have constants $0 < a < b$ and a function $r$ of time $t$ with differential identity:*

$$\dot{r} \geq ar^{s-1} - br^{2s-1} \tag{180}$$

*Furthermore, assume $0 < r_0$ and it always the case that $r \leq 1$.*

*Let $k = \frac{a}{b}$, and $T = \inf\{t \geq 0 : r \geq k^2\}$, then:*

$$T \leq \frac{1}{bk^2} \left( \frac{2k}{r_0^{s-1}} + \log \frac{1}{1-k} \right) \tag{181}$$

*Proof.* If $r_0 \geq k^2$, then $T = 0$ and the bound is obviously true. So assume $r_0 < k^2 \leq k^{\frac{1}{s-1}}$, where the second inequality follows from the facts that $k < 1$ and $s \geq 2$.

Consider the change of variables $y = r^{s-1}/k$:

$$\dot{y} = \frac{1}{k}(s-1)r^{s-2}\dot{r} \tag{182}$$

$$\geq \frac{1}{k}(s-1)(ar^{2s-3} - br^{3s-3}) \tag{183}$$

$$\geq \frac{1}{k}(ar^{2s-2} - br^{3s-3}) \tag{184}$$

$$= \frac{b}{k}(kr^{2s-2} - r^{3s-3}) \tag{185}$$

$$= \frac{b}{k}(k^3 y^2 - k^3 y^3) \tag{186}$$

$$= bk^2 y^2 (1 - y) \tag{187}$$

For $t \in [0, T)$, the RHS will always be positive, so we can write

$$\frac{\dot{y}}{y^2(1-y)} \geq bk^2 \tag{188}$$

Simple algebra lets us rewrite:

$$\frac{\dot{y}}{y} + \frac{\dot{y}}{y^2} + \frac{\dot{y}}{1-y} \geq bk^2 \tag{189}$$

And integrating from $0$ to $t$ gives

$$\log y_t - \log y_0 - \frac{1}{y_t} + \frac{1}{y_0} - \log(1 - y_t) + \log(1 - y_0) \geq bk^2 t \tag{190}$$

Remind that $\frac{1}{y_t} > 0$ and collecting terms, we have:

$$-\log\left(\frac{1}{y_t} - 1\right) + \log\left(\frac{1}{y_0} - 1\right) \geq bk^2 t - \frac{1}{y_0} \tag{191}$$

Taking exponentials and simple bounds:

$$\frac{1}{y_t} - 1 \leq \frac{1}{y_0} \exp\left(-bk^2 t + \frac{1}{y_0}\right) \tag{192}$$

Rearranging and reminding $y_t = r_t^{s-1}/k$

$$\frac{k}{1 + \frac{1}{y_0} \exp\left(-bk^2 t + \frac{1}{y_0}\right)} \leq r_t^{s-1} \leq r_t \tag{193}$$

To finish the proof, we'll show that $r_t \geq k^2$ is implied by a condition on $t$. Suppose that

$$t \geq \frac{1}{bk^2}\left(\frac{2}{y_0} + \log\frac{1}{1-k}\right) \tag{194}$$

Then using the fact that $k < 1$, and $\log x < x$ for all $x > 0$, it follows

$$t \geq \frac{1}{bk^2}\left(\frac{1}{y_0} + \log\frac{1}{y_0} + \log\frac{k}{1-k}\right) \tag{195}$$

$$\geq \frac{1}{bk^2}\left(\frac{1}{y_0} + \log\frac{1}{y_0\left(\frac{1}{k}-1\right)}\right) \tag{196}$$

Rearranging implies that

$$k \leq \frac{1}{1 + \frac{1}{y_0}\exp\left(-bk^2t + \frac{1}{y_0}\right)} \tag{197}$$

and plugging this into Equation 193 implies that $r_t \geq k^2$. Hence, the stopping time $T$ obeys:

$$T \leq \frac{1}{bk^2}\left(\frac{2}{y_0} + \log\frac{1}{1-k}\right) \tag{198}$$

Plugging in the definition of $y_0$ gives the bound. □

## H.2 Concentration Inequality Lemmas

We require a few very standard lemmas, adapting concentration inequalities to the complex setting.

**Lemma H.3.** *If $w$ is drawn from the standard complex Gaussian on $M$ dimensions, then*

$$P(|\|w\| - \sqrt{M}| \geq t) \leq 2\exp(-ct^2) \tag{199}$$

*for some universal constant $c$.*

*Proof.* Note that an equivalent way of sampling a complex Gaussian is $w = \frac{1}{\sqrt{2}}(w_R + iw_C)$ with $w_R, w_C$ both sampled iid from a standard real Gaussian on $M$ variables. Therefore

$$\|w\|^2 = \frac{\|w_R\|^2 + \|w_C\|^2}{2} \tag{200}$$

$$= \frac{1}{2}\left\|\begin{bmatrix} w_R \\ w_C \end{bmatrix}\right\|^2 \tag{201}$$

Note that $\hat{w} := \begin{bmatrix} w_R \\ w_C \end{bmatrix}$ is simply a standard Gaussian on $2M$ variables, so from Theorem 3.1.1 in Vershynin (2018):

$$P(|\|w\| - \sqrt{M}| \geq t) = P(|\|\hat{w}\| - \sqrt{2M}| \geq t\sqrt{2}) \tag{202}$$

$$\leq 2\exp(-ct^2) \tag{203}$$

for some universal constant $c$.

□

**Lemma H.4.** *Let $a_m \sim S^1$ be sampled iid, for $m = 1, \ldots, M$, and define $X \in \mathbb{C}^{N \times M}$ as $X_{nm} = a_m^n$. Then if we choose $M = O(N^3)$, with probability $1 - 2\exp(-\Omega(N))$:*

$$\sigma_1(X) = \Theta(\sqrt{M}), \sigma_N(X) = \Theta(\sqrt{M}) \tag{204}$$

*Proof.* Note that the columns of $X$ are independent, mean zero, and isotropic. Let $X_m$ be the $m$th column, and consider any $v \in \mathbb{C}^M$ with $\|v\| = 1$. Note that $\|X_m\| = \sqrt{N}$, so it follows that

$$\|\langle X_m, v \rangle\|_{\psi_2} \leq \sqrt{N} \tag{205}$$

where $\|\cdot\|_{\psi_2}$ denotes the subgaussian norm (Vershynin, 2018). Hence, we can apply Theorem 4.6.1 from Vershynin (2018) to $X^T$. Note, although this proof assumes real-valued variables, the same arguments follow through with no change to complex variables given the subgaussian bound on $\|\langle X_m, v \rangle\|_{\psi_2}$. Hence,

$$\sqrt{M} - cN(\sqrt{N} + t) \leq \sigma_N(X) \leq \sigma_1(X) \leq \sqrt{M} + CN(\sqrt{N} + t) \tag{206}$$

for universal constants $c, C$ and with probability $1 - 2\exp(-t^2)$. Choosing $t = \sqrt{N}$ and $M = O(N^3)$ gives the result.

$\square$

### H.3 VALID ACTIVATIONS

We quickly note a one simple choice of many possible activation functions that meets our criteria in Assumption 4.4.

**Proposition H.5.** *Let $\sigma(z) = \arctan \xi z + \xi z \arctan \xi z$ for $\xi = 1 - \frac{1}{\sqrt{N}}$. Then this activation satisfies Assumption 4.4, and $\sigma_+ \leq \sqrt{2}$, $\sigma_- \geq O\left(\frac{1}{N^{1/4}}\right)$.*

*Proof.* Observe that $\sigma$ is analytic on the unit disk following from properties of $\arctan$, with the Laurent series

$$\sigma(z) = \xi z + \xi^2 z^2 - \frac{\xi^3}{3} z^3 - \frac{\xi^4}{3} z^4 + \ldots \tag{207}$$

So the only coefficient equal to zero is the constant term. Moreover, if split into sequence of odd degree and even degree coefficients, both sequences are decreasing in absolute value, so we can instantly say that $\sigma_+ \leq \sqrt{2}$ and

$$\sigma_- = \min_{1 \leq k \leq \sqrt{N}} |c_k| \sqrt{k} \tag{208}$$

$$\geq \frac{\left(1 - \frac{1}{\sqrt{N}}\right)^{\sqrt{N}}}{\sqrt{N}} N^{1/4} = O\left(\frac{1}{N^{1/4}}\right) \tag{209}$$

Moreover, we can calculate:

$$\sum_{k=N+1}^{\infty} k|c_k|^2 \leq \sum_{k=N+1}^{\infty} \frac{k \xi^{k-1}}{(k-1)^2} \tag{210}$$

$$\leq \sum_{k=N+1}^{\infty} \xi^{k-1} \tag{211}$$

$$\leq \frac{\xi^N}{1 - \xi} \tag{212}$$

$$\leq e^{-\Omega(\sqrt{N})} \tag{213}$$

$\square$