# OpenReview forum: "Symmetric Single Index Learning"
_ICLR.cc/2024/Conference — ICLR 2024 poster_

### Official Review · Reviewer_7zCe · 2023-10-25

**Soundness:** 3 good
**Presentation:** 3 good
**Contribution:** 3 good
**Rating:** 6
**Confidence:** 4

**Summary:**

This paper studies the population gradient flow dynamics for learning a single index model using a symmetric neural network. It is shown that under powersum polynomial features and inputs following the Vandermonde distribution, identities appear that are similar to those of Hermite polynomials in Gaussian space. Using this property, the authors track the high-dimensional gradient flow dynamics using a small number of summary statistics and show the recovery of the hidden direction in the single index model.

**Strengths:**

This paper presents an interesting analysis that goes beyond the standard two-layer fully connected network and Gaussian input assumptions of the recent literature on single index learning, specifically by considering symmetric neural networks under a Vandermonde distribution. Furthermore, the authors are able to generalize the intuitions of Gaussian single index learning by showing that a proper notion of *information exponent* is still controlling the complexity of learning.

**Weaknesses:**

* There is no finite-sample analysis which is in contrast to the majority of recent works on learning single index models which had a particular focus on the sample complexity of the learning problem.

* Certain assumptions of the work are a bit restrictive. For example, the assumption on the activation is only verified for a particular choice that is not commonly used in practice, and the student link function $g$ needs to be carefully constructed using full information on the teacher link function $f$. However, these restrictions are understandable given that this work is a first attempt of studying the training dynamics of symmetric neural networks under a single index model.

* The writing of the paper can be improved with examples provided below.

**Questions:**

* The authors have mentioned why analyzing the SGD counterpart of the population gradient flow using the drift and martingale decomposition technique can be challenging. Another avenue for obtaining sample complexity is to use full-batch (empirical) gradient flow similar to Bietti et al., 2022 instead of the population version. I'm wondering if the six-stage proof can be adapted to handle such concentration errors and lead to a sample complexity.

* A particularly relevant paper could be [1] where the authors consider learning a single index model with anisotropic Gaussian data, and show that a structure in the covariance can reduce sample/runtime complexity and even remove dependency on the information exponent. I am wondering if similar observations can be made in this paper by introducing additional structure in the inputs. Furthermore, some aspects of the current analysis also appear in [1] due to dealing with anisotropy, such as having to control a quantity of the type $\Vert f(\langle Aw, x\rangle)\Vert_{L^2}^2$ or preconditioning, which might be interesting to point out.

* The squared Vandermonde density is never explicitly defined in the manuscript. Perhaps providing an explicit definition and recalling some of its relevant properties can help the readers better understand the problem setting.

## Minor Comments:
* Is the probability in Lemma 4.5 is over the randomness of $(a_m)$? In that case, it might be helpful to point this out in the statement of the lemma.
* Theorem 4.2 point (iii) asks for $v_0 = 1 - r_0^2$. Could it be more intuitive to introduce this condition as $\Vert Aw_0 \Vert = 1$?
---
[1] Alireza Mousavi-Hosseini, Denny Wu, Taiji Suzuki, Murat A. Erdogdu. "Gradient-Based Feature Learning under Structured Data." NeurIPS 2023.

---

> ### Author Response · Authors · 2023-11-23
>
> Thank you for your review, our response is below:
>
> We agree that a finite-sample analysis of full-batch gradient descent would be helpful, see our response to reviewer qFtS on that topic.
>
> We agree the restrictions are non-negligible, although we note that some are seemingly necessary: for instance, using a different link function demonstrably improves the rate of convergence, and the theoretical issue of trying to optimize a polynomial landscape if one used the correlational loss with the same link function for student and teacher necessitate the restriction.
>
> We agree that the results on learning anisotropic Gaussian data are related in the sense of needing to handle norms of individual single index terms, and will mention these connections in a later draft.  We also agree that giving a proper characterization of the Vandermonde density, and the suggestions to the Lemma and Theorem statements, are all worthwhile for clarity.

---

> > ### Comment · Reviewer_7zCe · 2023-11-23
> >
> > Thank you for your responses. I maintain my positive evaluation of the work.

---

### Official Review · Reviewer_qFtS · 2023-10-28

**Soundness:** 4 excellent
**Presentation:** 3 good
**Contribution:** 3 good
**Rating:** 8
**Confidence:** 3

**Summary:**

This paper studies the convergence guarantees of gradient flow for the problem of retrieving the unknown parameters of a symmetric single index model with a known analytic link function, under appropriate assumptions on the distribution of the features, which are complex vectors. It is shown that gradient flow on some appropriately defined loss function based on a DeepSets student model structure converges to a student model whose parameters are highly correlated with the ground truth, in time that scales logarithmically with the approximation error parameter and exponentially to the (square of the) information exponent associated with the link function, whenever the chosen student model satisfies certain properties. Moreover, an explicit choice of the student model is proposed and precise bounds for the associated convergence rate are provided. The authors also provide experimental demonstration of their results on synthetic data.

The results in this paper work under a set of modeling assumptions. First, the teacher model is assumed to have a symmetric structure, in the sense that variable relabelling does not change the value of the function. Each of the unknown parameters influences each variable in the exact same way and, in particular, the $k$-th parameter multiplies the sum of all the monomials of degree $k$. Moreover, the link function is assumed to be analytic and supported on monomials of degree at most equal to the square root of the input dimension. Finally, the input distribution is assumed to be the squared Vandermode density, which is shown to enjoy properties in the symmetric setting analogous to those of the Gaussian for a non-symmetric setting.

The proof techniques used include complex calculus as well as a crucial identity of Vandermode distributions on the symmetric polynomials setting, which motivates establishing Vandermode distributions as the Gaussian analogue in the symmetric polynomials setting.

**Strengths:**

The paper provides the first positive result in the proposed setting and acquires bounds in terms of the information exponent, which is well-studied quantity. The presentation is clear, sufficiently detailed and accurate. The techniques proposed in this paper might be of independent interest, especially as they motivate further study of learning problems under Vandermode marginals.

**Weaknesses:**

As the authors mention in their limitations section, the (distributional and modeling) assumptions required for the proposed analysis to work are not as common in the literature. Therefore, it is not clear to what extent such assumptions are realistic or significantly valuable from a theoretical perspective. That said, this work provides results that apply to a very large class of link functions, which partly justifies a certain number of strong distributional assumptions.

Another weakness of this work is that the results do not seem to be immediately interpretable in the Probably Approximately Correct (PAC) learning setting, since it is implicitly assumed that one has access to exact gradient oracles for optimizing the chosen loss function.

**Questions:**

My main question concerns my second comment in the weaknesses section: Can the results provided in this paper be translated to standard PAC learning guarantees? If not, then what are the main obstacles in doing so?

---

> ### Author Response · Authors · 2023-11-23
>
> Thank you for your review, our response is below:
>
> We agree that the analytic assumption is a strong one; our main goal is to show even a limited setting where learning permutation invariant models is provable.
>
> We agree that a future work would hopefully be able to show the result still holds for, say, full gradient descent, as we're not aware of a conceptual reason why concentration of the gradient wouldn't give the same evolution as gradient flow.  Stochastic gradient descent is likely much harder: the main issue is phase 1 in the ODE convergence proof, where we need to show that log r^2/v is decreasing in order to guarantee that the magnitude of the convergence r doesn't become unacceptably small.  It seems potentially possible to adapt the submartinagle arguments of [1] by taylor expanding the logarithm and defining appropriate stopping times, but it would require multiple expansions since r^2/v can vary dramatically in magnitude and different expansions would give different rates of convergence, so somewhat hard to tell if the rates would still be polynomial overall or not.

---

> > ### Comment · Reviewer_qFtS · 2023-11-23
> >
> > Thank you for your response. My score remains the same.

---

### Official Review · Reviewer_oXjh · 2023-11-09

**Soundness:** 4 excellent
**Presentation:** 4 excellent
**Contribution:** 2 fair
**Rating:** 5
**Confidence:** 4

**Summary:**

This paper studies the problem of studying single-index functions on the feature space given by power-sum polynomials.
Thus, all of the teacher functions fall into the class of symmetric functions, which are those functions such that f(x1,...,xn).

Under a specific data distribution that allows for an inner product with nice orthogonality properties, this paper analyzes the gradient flow of learning with a student model under the correlation loss.

An information exponent (similar to Ben arous et al.), plus the initial correlation with the ground truth, is shown to upper-bound the time needed to learn.

**Strengths:**

The presentation is clear and easy to follow.

The problem of provable guarantees for the dynamics of learning symmetric functions has not been studied as far as I know, so this can be of interest to the community.

Related literature is covered well.

The analysis seems correct.

**Weaknesses:**

1) "These dynamics naturally motivate the question of learning efficiency, measured in convergence rates in time in the
case of gradient flow". Does this say anything about sample complexity when discretized? Because gradient flow time can be rescaled, so it doesn't appear to be a well-defined complexity measure?

2) Can you prove a converse to Theorem 4.2 with corresponding lower bounds on the time? This seems doable and like it would strengthen the result to make it more of a characterization.

3) It is unclear to me what is conceptually new in this work that does not appear in previous analyses of single-index learning?
* (a) the teacher model / student model are single-index models, but on a different data distribution than usual. But this has a similar inner product to what allows analyzing the Gaussian case.
* (b) the analysis of the single-index model appears to follow a now-standardized template. Would be interesting for the authors to highlight what are the new elements, or how this generalizes the currently-known techniques.

**Questions:**

See questions in the weaknesses section.
Also:
4) It is unclear to me what I am supposed to be learning from the experiments in Figure 1. How does the initial alignment of the model correspond to the gradient flow time in these cases?

Typos:
* "The former assumptions essentially corresponds"
* "any other works that demonstrates"
* "dynamcis"
* Assumption 4.4(iv) seems like it should be Omega(sqrt(N)) instead of O(sqrt(N)); Lemma 4.5 should be exp(-Omega(N)) instead of exp(-O(N)), and similarly for Lemma 4.6; other places including in the appendix there are some Omega(.) vs O(.) issues

---

> ### Author Response · Authors · 2023-11-23
>
> Thank you for your review, our response is below:
>
> Although gradient flow can be rescaled, we observe evidence from the literature that using constant rescaling is comparable to the sample complexity from comparable settings in gradient descent or stochastic gradient.  For example, in Theorem 5.3 in [1] the rates for all three methods are essentially equivalent.
>
> In terms of lower bounds, it would be straightforward to have a lower bound of N^s.  Indeed, the convergence can only be faster if we initialize v = 0 and cos s theta = 1, in which case the ODE governing r is nearly equivalent to the one studied in [2].  However, unlike [2], it's not straightforward to get nearly tight upper and lower bounds on the problem: for one thing, the initialization guarantees are based on the condition number of the frozen matrix A, which is essentially equivalent to the condition number of a rectangular Vandermonde with nodes on the unit complex circle.  As far as we're aware this is still an active area of research to get tight bounds here, one representative paper in [3].
>
> In terms of novelty, although other papers that use correlation loss [4], they can still track learning via a single parameter.  We necessarily have three parameters, since our correlation is complex and we can't project without solving the optimization problem onto an ellipsoid [5] at every step.  Therefore the technical novelty is treating the learning with a system of three ODEs, and in particular judiciously choosing the regularization so that the orthogonal part of the learned direction (v) shrinks faster than the correlation with the true direction (r) to guarantee eventual convergence.
>
> The experiments are to confirm the expected behavior still occurs under full gradient descent, and to observe that the technical details of controlling the descent of r are not a proof artifact but a necessary detail, as the regularization pushes r down in the initlal parts of training.  As in the vanilla single index setting, higher initial alignment usually corresponds to faster convergence.
>
> [1] Jia, Zehui, Xingju Cai, and Deren Han. "Comparison of several fast algorithms for projection onto an ellipsoid." Journal of Computational and Applied Mathematics 319 (2017): 320-337.
>
> [1] Yehudai, Gilad, and Shamir Ohad. "Learning a single neuron with gradient methods." Conference on Learning Theory. PMLR, 2020.
>
> [2] Arous, Gerard Ben, Reza Gheissari, and Aukosh Jagannath. "Online stochastic gradient descent on non-convex losses from high-dimensional inference." The Journal of Machine Learning Research 22.1 (2021): 4788-4838.
>
> [3] Kunis, Stefan, and Dominik Nagel. "On the condition number of Vandermonde matrices with pairs of nearly-colliding nodes." Numerical Algorithms 87 (2021): 473-496.
>
> [4] Damian, Alex, et al. "Smoothing the Landscape Boosts the Signal for SGD: Optimal Sample Complexity for Learning Single Index Models." arXiv preprint arXiv:2305.10633 (2023).
>
> [5] Jia, Zehui, Xingju Cai, and Deren Han. "Comparison of several fast algorithms for projection onto an ellipsoid." Journal of Computational and Applied Mathematics 319 (2017): 320-337.

---

### Official Review · Reviewer_xBy6 · 2023-11-10

**Soundness:** 3 good
**Presentation:** 3 good
**Contribution:** 2 fair
**Rating:** 6
**Confidence:** 2

**Summary:**

The paper analyzes single-index learning, especially when the underlying teacher model has permutation invariance and the student model is a variant of the DeepSets architecture.
The paper shows that under the squared Vandermonde density over a complex domain, the hidden direction (of a finite support in the power-sum polynomials feature space) can be recovered up to an $s$-th root of unity via the gradient flow applied for a variant of correlation loss with regularization loss, where $s$ is the information exponent. A provably good initialization scheme is also provided as well as a concrete form of a good activation function that meets the regularity condition.

**Strengths:**

- Understanding convergence of a neural network training with gradient-based methods is an important problem, and this paper provides a new theoretical framework that admits a clean analysis. The problem setting and techniques may be useful for further work in this line of study.
- The paper is well written overall. I am new to this area, but I was able to understand a high-level landscape of the previous works and the contribution of this work compared to the existing works. Though there are several simplifying assumptions on the teacher network, the input distribution, the loss function, and etc., the rationale behind them are clearly explained and discussed appropriately. The reasonings behind the choice of loss functions and the assumptions are well thought out and helpful.
- A technical difficulty in a SGD analysis is also insightful.

**Weaknesses:**

- There are several assumptions that limit the applicability of the framework, though they are properly discussed. Proposition 2.3 is the key technique that governs the whole technical assumptions including the assumption on the link function (Assumption 2.4). It is unclear for me as a newbie reader to this field how much it is restrictive and whether this can be relaxed. Any further discussion would be appreciated.
- Though the paper starts with the single-index learning, it seems that the framework deviates from the framework by considering a different link function in the final form of the modified correlation loss in eq. (14). I found it quite confusing at first sight. After this modification, is this technically qualified as a single-index learning?

**Questions:**

Minor suggestions
- After eq. (6), "In other words" sounds confusing. It might be better to explicitly say that the first-layer weights ($\{a_1,\ldots,a_M\}$) and the third-layer weight (the function $g$) will be fixed (frozen), and the second-layer weight ($w$) will be the only trainable parameter.
- In Corollary 4.7, delete "And" before "Consider".
- Please consider different notation for the summary statistics in Theorem 4.1 other than $m$ as it clashes with its other usage as an index.
- The usage of $\inf$ in the definition of the information exponent seems unnecessary.
- Figure 1 can be much improved. For examples, y-axes can be shifted to hide unused range.

---

> ### Author Response · Authors · 2023-11-23
>
> Thank you for your review, our response is below:
>
> The assumptions on the link function can be slightly relaxed, in the sense that allowing for very small terms of higher degree would have minimal effect on the gradient overall.  But the analytic assumption is likely required due to the fragile nature of the hermite-like identity.  We will include more discussion of this in an updated draft.
>
> This is certainly still a single-index model in the sense of trying to recover a single hidden direction that can only be accessed through non-linear measurements.  And there is precedent for using a different link function in the student architecture than the teacher in the vanilla single index setting, see [1].
>
> The suggestions given for clarity are appreciated.
>
> [1] Dudeja, Rishabh, and Daniel Hsu. "Learning single-index models in gaussian space." Conference On Learning Theory. PMLR, 2018.

---

> > ### Comment · Reviewer_xBy6 · 2023-11-23
> >
> > I appreciate the authors' response. I will keep my score.

---

### Meta-Review · Area_Chair_sRrq · 2023-12-13

**Metareview:**

Authors consider learning single index models, which are statistical models where the response variable depends on the input through a direction determined by a single unit vector.  They use a symmetric neural network under powersum polynomial features.
The inputs are assumed to follow Vandermonde distribution. The paper consider the gradient flow dynamics and prove that it can learn the single index direction using a few summary statistics.

This paper was reviewed by 4 reviewers and received the following Rating/Confidence scores: 5/4, 6/2, 8/3, 6/4.

I think the paper is overall interesting and should be included in ICLR. The authors should carefully go over and address all reviewers' suggestions.

**Justification For Why Not Higher Score:**

The assumptions and the setting is a little limited.

**Justification For Why Not Lower Score:**

n/a

---

### Decision · Program_Chairs · 2024-01-16

Accept (poster)